# Recent Advances in Isolated Single-Atom Catalysts for Zinc Air Batteries: A Focus Review

**DOI:** 10.3390/nano9101402

**Published:** 2019-10-02

**Authors:** Weimin Zhang, Yuqing Liu, Lipeng Zhang, Jun Chen

**Affiliations:** 1School of Chemistry and Chemical Engineering, Shandong University of Technology, Zibo 255049, China; wmzhang@sdut.edu.cn; 2ARC Centre of Excellence for Electromaterials Science, Intelligent Polymer Research Institute, Australian Institute of Innovative Materials, Innovation Campus, University of Wollongong, Wollongong, NSW 2522, Australia; yuqing@uow.edu.au

**Keywords:** Single-atom catalysis, zinc–air battery, oxygen reduction

## Abstract

Recently, zinc–air batteries (ZABs) have been receiving attention due to their theoretically high energy density, excellent safety, and the abundance of zinc resources. Typically, the performance of the zinc air batteries is determined by two catalytic reactions on the cathode—the oxygen reduction reaction (ORR) and the oxygen evolution reaction (OER). Therefore, intensive effort has been devoted to explore high performance electrocatalysts with desired morphology, size, and composition. Among them, single-atom catalysts (SACs) have emerged as attractive and unique systems because of their high electrocatalytic activity, good durability, and 100% active atom utilization. In this review, we mainly focus on the advance application of SACs in zinc air batteries in recent years. Firstly, SACs are briefly compared with catalysts in other scales (i.e., micro- and nano-materials). A main emphasis is then focused on synthesis and electrocatalytic activity as well as the underlying mechanisms for mono- and dual-metal-based SACs in zinc air batteries catalysis. Finally, a prospect is provided that is expected to guide the rational design and synthesis of SACs for zinc air batteries.

## 1. Introduction

### 1.1. Why ZABs?

Nowadays, based on the current advanced technology, the energy density of lithium ion batteries can reach only approximately 100~250 Wh kg^−1^. Therefore, it is urgent and critical to develop alternative chemical power sources to satisfy the increasing demands of various portable and stational electric devices. Among various newly developed post-lithium-ion battery technologies, metal–air batteries have the potential to provide several times higher performance than the current advanced lithium ion batteries (in terms of gravimetric energy density) and have therefore been regarded as promising candidates to power various electric devices.

Typically, metal–air batteries are fuel cells comprised of a metal anode (like Li, Na, K, Mg, Al, Zn, and Fe), an air electrode, a proper amount of electrolytes, and a separator [1,2,3,4,5,6,7,8,9,10]. In terms of full cell voltage, the alkali metal–air batteries have the highest values of 2.96, 2.33, and 2.48 V for Li–air, Na–air, and K–air batteries, respectively, which are higher than those other metal–air batteries. The theoretical energy density for diverse metal–air batteries and the Ragone plots for some important and mostly investigated metal–air batteries are given in Figure 1. We can see that among the various metal–air batteries, lithium–air batteries possess a high theoretical specific mass energy density, up to 11,430 Wh kg^−1^ (excluding oxygen uptake). However, due to the low mass densities of the alkali metals, the corresponding volume energy densities of Li–air, Na–air, and K–air are 6104, 2634, and 1513 Wh L^−1^, respectively, which are lower than zinc–air batteries (9653 Wh L^−1^) and other metal–air batteries. [11] In addition, their serve hazard issues significantly limit their practical applications [10]. For other metal–air batteries, Al can be easily dissolved in either alkaline or acid electrolytes, which greatly impedes the application of aluminum–air batteries. Mg–air and Al–air batteries suffer from rapid self-discharge via the hydrogen evolution reaction due to the low standard electrode potentials of Mg (−2.372 V vs. Reversible Hydrogen Electrode (RHE)) and Al (−1.662 V vs. RHE). In addition, both of them are not electrically rechargeable. Compared to the above-mentioned metal–air batteries, zinc–air batteries exhibit a high specific capability, a high theoretical high energy, excellent safety, high corrosion resistance and an earth abundance of zinc metal anode. These merits probably make them the most practically viable battery systems.

### 1.2. Working Principle of ZABs

There are three main types of zinc–air batteries (ZABs), namely primary, secondary and mechanically rechargeable batteries. A primary ZAB is applied for single use and is then discarded after the metal anode is completely consumed. A secondary rechargeable battery system is returned to its original state by forcing current in the reverse direction versus that of the discharge. In a mechanically rechargeable ZAB, the depleted anode and failure electrolyte are replaced to repeatedly recover the capacity for the battery, so mechanically rechargeable ZABs are a class of refurbishable primary cells.

According to the type of electrolyte, ZABs can be categorized into aqueous and non-aqueous types. The corresponding electrolytes are alkaline solutions and solid polymer electrolytes (SPEs) or room temperature ionic liquids (RTILs). Among them, aqueous ZABs (especially those using the KOH electrolyte) are the most mature and reliable battery system [12,13]. In an alkaline primary ZAB, as shown in Figure 2a, the zinc metal is oxidized at the air electrode with an electron release during the discharge process. The resulting Zn^2+^ further reacts with the OH^−^ diffused from the cathode, which generates soluble zincate ions (Zn(OH)_4_^2−^). The zincate ions are increasingly supersaturated upon the battery discharge and then decomposed into insoluble ZnO. The electrons pass through the electric circuit and provide electric energy to various electric devices. Simultaneously, O_2_ accepts the electrons generated from the anode, and the oxygen reduction reaction (ORR) occurs, which yields OH^−^. Since zinc is an active metal and self-corrosion is unavoidable, Zn(OH)_2_ and H_2_ are yieled. Typically, the detailed reactions in the discharge process for an alkaline ZAB are summarized as the follows.

Anode: Zn + 4OH^−^ → Zn(OH)_4_^2−^ + 2e^−^(1)

Zn(OH)_4_^2−^ → ZnO + H_2_O + 2OH^−^(2)

Cathode: O_2_ + 2 H_2_O + 4e^−^ → 4OH^−^(3)

Total reaction: 2Zn + O_2_ → 2ZnO(4)

Parasitic reaction: Zn + 2H_2_O → Zn(OH)_2_ + H_2_(5)

In terms of the electrochemical reactions, both primary and mechanically rechargeable ZABs operate only in a discharge mode which involves the above-mentioned electrochemical reactions. For a secondary rechargeable ZAB, however, a charge process is also involved which is fulfilled by reversing the reactions (Figure 2b). The corresponding reactions include the oxygen evolution reaction (OER) at the air electrode and the zinc metal plating at the negative electrode.

### 1.3. Challenges and Opportunities

In recent years, ZABs have been receiving much attention and achieving great progress. It is worth mentioning that primary ZABs have been commercially implemented for telecommunication and medical applications. However, despite their primary commercialization and promising potential, the development of ZABs has been impeded by the main issues associated with the metal and air electrodes. In addition, issues including non-uniform zinc dissolution and deposition, as well as the lack of a satisfactory bifunctional electrocatalyst, have greatly impeded the cyclability of secondary ZABs. Despite the high theoretical voltage of 1.66 V created by the thermodynamically spontaneous cathodic and anodic reactions, the practical operation voltage is normally less than 1.2 V. Moreover, to re-charge a ZAB, a much higher voltage of 2 V is often required to reverse the reactions. This voltage is mainly raised by the large electrode polarizations of the ORR and the OER at the air electrode and the parasitic oxidation of the Zn metal, as well as its dendritic growth at the Zn electrode. To improve the ORR/OER performance and maximum the energy output efficiency as well as the long-term cyclability for ZABs, effort has been continuously dedicated to exploring highly active electrocatalysts for their air electrode reactions. Thus far, many electrocatalyst including precious metals and alloys [14,15,16], hetero-atoms doped carbonaceous materials [17,18,19,20], transition metals and their carbides and oxides [21,22,23], and perovskite materials [24] have been developed. For more efficient and enhanced electrocatalysts, common strategies include tuning the particle size, morphology, crystalline facets, electronic structure, and hybridization to a rational state [25,26,27,28]. For perovskite materials, anion-deficient tuning is an effective strategy to enhance their ionic conductivity, which may further improve their ORR/OER performance [29,30,31,32]. Among all strategies, downsizing the catalyst particle is the most effective and straightforward one to increase the catalytic active sites while also increasing the under-coordinated metal atoms sites. Theoretically, a maximum utilization of 100% of metal atoms can be reached when the size is reduced to an atomic scale, which corresponds to single atom catalysts (SACs). In the past several years, SACs have attracted much attention in catalysis science due to their boosted catalytic activity, selectivity, stability, and the reduced utilization of their active metals [33,34,35,36,37,38,39].

In this review, we limit the scope to the recent advances of SACs for ZABs. Emphasis is focused on the discussion of the synthetic methodologies and the electrocatalytic activity as well as the underlying mechanisms of SACs for the ORR/OER in ZABs.

## 2. SACs in ZABs

SACs are materials with isolated atomic metal active sites dispersed on a support at an atomic state. The concepts of single-atom catalysis and single-atom catalyst were pioneered by Zhang and co-workers in 2011. They synthesized single Pt atoms on FeO_x_ for the catalysis of CO oxidation. Due to the alteration of the electronic structure of Pt and the charge transfer from Pt to FeO_x_, the single-atom catalyst exhibited exceptional catalytic activity and stability [36]. A single-atom catalyst possesses many distinguished properties which are different from those of its nanostructured or sub-nanostructured countparts. For example, when the dispersion level of the atoms in the catalyst reach atom scale, there is a sharp increase in surface apparent energy, quantum size effect, and unsaturated coordination environment, as well as a different interaction between the metal and the support. Due to the fact that only the coordinately unsaturated atoms on the surface of catalysts can be involved in the catalytic process, all these resulted characters endow the catalyst with superior catalytic performance compared with their bulky and nanostructured counterparts. In addition, according to Figure 3, SACs inherit the merits of both heterogeneous and homogeneous catalysts. In catalyzing the ORR/OER in ZABs, it is expected that single-atom catalysis exhibits the potential to make best use of the advantages while also bypassing the disadvantages of heterogeneous and homogenous catalysis [32,33,34,35].

The most extensively studied and effective SACs for ZABs are M–N_x_/C (M = Fe, Co, Mn, Cu, etc.; x = 2, 4, 6, etc.) supported on a carbonaceous matrix. The M_x_–N/C catalysts are usually those with the isolated active metal centers dispersed on a support and stabilized by covalent coordination or the interaction of neighboring surface atoms such as nitrogen, oxygen or sulfur. The average order of the ORR/OER activity of the MN_x_-based SACs is Fe–N_x_/C > Co–N_x_/C > Mn–N_x_/C > Cu–N_x_/C [44,45,46,47,48,49,50,51,52,53,54,55]. The low cost M–N_x_/C catalysts that use diverse transition metals as their centers exhibit outstanding catalytic activity for the ORR/OER in ZABs and have been regarded as alternatives to Pt-based catalysts [50,51,52,56,57]. Though there is no M–M bond to break the O–O bond for M–N_x_ moiety-based SACs, M–N_x_–C-based catalysts can promote the ORR with a high performance through a four-electron pathway. It has been suggested that these catalysts’ highly dispersed as well as coordinatively unsaturated environment and enhanced charge-transfer effect by the interaction of the metal-support remarkably increase catalytic active sites. Their unique electronic structures also result in the appropriate adsorption energies for oxygen and thus decrease the free energies, all of which eventually leads to improved intrinsic catalytic activity and the four-electron pathway for the ORR [58,59]. In addition, owing to the fact that metal centers are confined by heteroatoms in the carbon matrix, their electrocatalytic stability is significantly enhanced. Thus far, among the diverse non-noble metals, Fe and Co are the most commonly employed metals used to construct M–N_x_ moiety sites and have been proven to perform best for various electrochemical catalytic reactions [37,60,61,62,63,64,65].

## 3. Advances of SACs in ZABs

### 3.1. Fe-N_x_/C-Based SACs

#### 3.1.1. Mono-Metal Fe-N_x_/C-Based SACs

Generally, it has been established that Fe–N_x_/C intrinsically perform best for the ORR among the various N–M_x_/C SACs because of the stronger adsorption of O_2_ on the Fe–N_x_ moiety than other moieties [59]. To synthesize Fe–N_x_/C, previous synthesis methodologies usually use pyrolyzing iron-based macrocycle complexes or a simply mixed precursor containing iron and a nitrogen source. These methods often involve a wet impregnation procedure and a subsequent heat treatment at temperatures ranging from 650 to 1000 °C in an inert atmosphere [66]. One of the drawbacks is that this is likely to result in the formation of inert metal particles and oxides or other less active species. Therefore, metal loading must be kept <1 wt% to avoid aggregation during synthesis [53]. Due to the lack of sufficient active sites, the catalytic activity is quite low. For example, ZABs with a low loading of a 0.26% Fe–N_x_/C catalyst can only deliver a maximum power density of ~70 mW cm^−2^ and a specific capacity of 705 mAh g^−1^ at 5 mA cm^−2^ [53].

The most effective approach to enhance the performance of SACs is probably to increase the density of active sites on the support while promoting the intrinsic activity of the active sites. To increase the number of active sites, the well-established and most often employed synthetic methodologies are based on defect engineering, spatial confinement strategies, and coordination design strategies [67]. When these strategies are rationally used in synthesis, a much higher metal loading may be obtained. For example, Yang et al. used a versatile molecule-confined pyrolysis strategy based on the wet-chemical method and successfully synthesized a single Fe atom-based SAC (Fe-SAs/N–C) with a high metal loading up to 3.5 wt% on a porous N-doped carbon matrix. In their study, they first confined the Fe^2+^ by coordinating it to 1,10-phenanthroline (phen) ligand and then using this coordinated system as the precursor. Due to the space isolation effect of the phen, the subsequent pyrolysis resulted in single Fe atoms rather than Fe nanoparticles. The as-synthesized catalyst exhibited a better electrochemical performance that of a commercial Pt/C catalyst for the ORR in a 0.1 M KOH aqueous electrolyte. In this work, they clarified that the mechanism of the catalytic activity is mainly derived from isolated atomic Fe–N_x_ moiety sites other than C–N_x_. Benefiting from this, a three-cell primary zinc–air battery stacked with an Fe–N_4_/C catalyst delivered a high maximum power density of 225 mW cm^−2^, and the specific capacity was ~636 mAh g^−1^ (Figure 4) [68].

Additionally, to obtain a high loading SAC, metal–organic framework (MOF)-based materials have been employed to assist the spatial confinement strategy. MOFs exhibit well-defined porous structures which can act as the spatial separation and confinement of the single metal atom loading. The highly ordered arranged organic linkers can be readily converted into a functional carbon support, and the metal species can be changed into a single metal-based active site after pyrolysis. Therefore, MOFs have great potential in synthesizing diverse metal-based SACs for the ORR/OER [46,69,70,71,72].

#### 3.1.2. Dual-Metal Fe–N_x_/C-Based SACs

Recent reports have indicated that introducing a different metal atom into Co or Fe SACs can more effectively facilitate the cleavage of oxygen–oxygen bonds to achieve a four-electron pathway during the ORR and to feature enhanced catalytic activity [73,74,75]. In such a structure, the center metal is coordinated with a foreign metal atom instead of being coordinated with nitrogen in the common simple single atom, leading to a structure such as Fe–Co and Zn–Co. For example, in the an Fe–Co system (Figure 5), the strong binding of an oxygen molecule on the dual site enables the sufficient activation of O–O from 1.23 to 1.40 Å, and the Fe–Co site provides two anchoring sites for the dissociated O atoms. As a result, the dissociation barrier of O_2_ and OOH into O and OH is much lower than that of the corresponding single sites catalysts [73].

In addition to the capability to catalyze the ORR, Fe–N_x_/C can also catalyze the OER. It has been thought that Fe–Nx/C is a highly active catalyst for the ORR, while much less study has been conducted in the field of the OER. Recently, Chen at al. synthesized an Fe–N_x_ on a two-dimensional (2D) highly graphitic-porous, nitrogen-doped carbon layer (FeN_x_–PNC). The FeN_x_–PNC not only exhibits a good catalytic activity for the ORR but also shows exceptional OER performance with a low overpotential of 390 mV at 10 mA cm^−2^. This character demonstrates the promising potential of Fe–N_x_/C for secondary rechargeable ZABs.

The common structure of Fe–N_x_/C is iron–nitrogen coordination dispersed on a conductive carbon support. Besides the rational design of the conductive structure, the bi-functional activity of the Fe–N_x_/C-based SACs can also be effectively enhanced by introducing other heteroatoms (S, P, B, O, etc.) into the carbon matrix (Figure 6) [47,51,76]. Besides nitrogen, the sulfur atom is probably the most widely used atom for doping strategies to enhance ORR/OER activity [47,77]. The presence of S and N dopants can result in an uneven charge distribution in the carbon framework which further leads to positively charged carbon atoms, thus facilitating the adsorption of oxygen species [77]. When B is used as a dopant atom in a carbon matrix, it can provide electron-deficient sites to improve the electron transfer for the Fe–N_x_–C sites and also strengthen the interaction with oxygen-containing species. A P dopant helps modify the electronic structure of Fe centers and weakens the adsorption of the *OH intermediate, which leads to improvement of the ORR. Meanwhile, the OER is also greatly enhanced [78].

In comparison with their mono-metal SACs counterparts, dual-metal Fe–N_x_/C-based SACs may also have a better OER activity. A recent work related to dual-metal (Fe and Ni) SACs was conducted by Liu et al. In this work, they claimed that the electronic structures of SACs are reconfigured after the introduction of another metal. In the case of an Fe/Ni dual-metal SAC, the presence of Fe notably increases the oxidative state of the Ni site, which can lead to improved OER activity for the catalyst [79]. This study may provide promising guidance for the design of dual-metal SACs for the bi-functionally catalyzation of the ORR/OER.

### 3.2. Co–N_x_/C-Based SACs

#### 3.2.1. Mono-Metal Co–N_x_/C-Based SACs

Apart from the Fe–N_x_/C catalysts, Co–N_x_/C is another widely investigated and effective catalyst for the ORR and the OER [51,76,80]. According to recent reports, despite its inferior ORR activity to Fe–N_x_/C, Co–N_x_/C displays highly active performance in catalyzing both the ORR and the OER. Co–N_x_/C was widely investigated for the ORR before the concept of “single-atom catalysis” proposed by Zhang and co-workers [66]. The Co–N_x_ catalysis system can be traced back to a study by Jasinski in 1964 who firstly reported the nitrogen–metal coordination (cobalt phthalocyanine) catalytic structure [81], whereas the catalyst displayed very poor stability. To improve the performance, a further pyrolysis procedure in an inert atmosphere which can convert the metal-macrocyclic complex into metal–N_x_ structure and effectively enhance the catalytic activity and stability for the ORR was employed [82]. From then on, extensive studies have been conducted on the development of various metal–N_x_ electrocatalysts [83,84,85,86,87]. These catalysts may be regarded as the primary SAC systems despite the fact that the concept was not proposed until in 2011 [36]. Similar to Fe–N_x_/C, mono-metal Co–N_x_/C can achieve a better performance than noble metal-based nanocatalysts in catalyzing the ORR/OER in alkaline aqueous-based ZABs [88,89]. A further improvement on the catalytic activity also requires increasing the active site number and promoting the intrinsic activity of each active site, strategies of which are similar to the aforementioned methods for Fe–N_x_/C.

#### 3.2.2. Dual-Metal Co–N_x_/C-Based SACs

For zinc–air batteries use, Zhao and co-workers designed a Zn–Co system coordinated on an N-doped carbon matrix (Zn/CoN–C) by a wet-chemical synthetic method. Both Zn and Co are coordinated N on a carbon matrix, and there is also a Zn–Co bond in the system. Density functional Theory (DFT) calculations have shown that O–O (1.23 Å) is slightly lengthened to 1.29 on ZnN_4_ and 1.30 Å on CoN_4_, whereas it is greatly increased to 1.43 Å on Zn/CoN–C. It has been indicated that ZnN_4_ is less active than CoN_4_ and Zn/CoN–C, while the coordination of dual-metal-atoms with nitrogen atoms is more favorable for the cleavage of O–O bond. This has been further confirmed by the electrochemical measurement and zinc–air battery performance evaluation. An in situ X-ray absorption near edge structure (XANES) analysis was carried out to further investigate the Zn/CoN–C structure and reaction mechanism, which suggests that Co is actually the active center during the ORR. Meanwhile, it has been shown that ZnN–C exhibits rather poor ORR electrocatalytic activity in an alkaline electrolyte (Figure 7). Therefore, it is reasonable to conclude that Zn/CoN–C can be regarded as a Zn-enhanced Co single-atom catalyst rather than a precise single atom catalyst in terms of structure and the electrochemical performance [74].

In addition, further doping the dual metal system with S atoms in the carbon layer can significantly enhance the intermediate adsorption and the electron transfer between the catalyst and the reaction intermediates generated in the ORR. The as-fabricated ZAB can deliver an improved power density [75].

### 3.3. Other Metal-Based SACs

Very recently, other transition metal- (such as Cu and Mn) based SACs have been exploited and developed for catalyzing the ORR in ZABs [54,55]. It has been generally accepted that Cu is a poorly active metal for the ORR due to its filled d-electron shell. Cu has only one s-orbital electron on top of a filled d-electron shell (3d^10^4s^1^), which impedes the electron interactions with oxygen-containing intermediates such as *OOH and *OH. This eventually weakens the oxygen chemisorption strength and thus leads to sluggish ORR kinetics [54,90]. However, when the size of Cu is reduced to the atomic scale and coordinated with the surrounding N atoms to form CuN_2_ or CuN_4_ (Figure 8a,b), the d electronic states are tuned to near the Fermi level; thus, the ORR catalytic activity is activated. This can be ascribed to the migration of the valence electrons from the Cu atoms to nitrogen. That is to say, atomically dispersing Cu can lead to a low coordination environment and an enhanced charge-transfer effect, which yields an increased number of active sites with enhanced intrinsic activity. Inspired by the catalytic activity of the ORR generated by the unsaturared Cu ion in Cu^2+^ enzymes, Yang et al. confirmed the isolated Cu single-atom catalyst exceeded Pt/C in ORR and zinc–air battery operations. The atomically dispersed Cu–N_x_ are dominantly Cu–eN_4_ with lesser amounts of Cu–N_2_. Typically, the ORR occurs on the two moieties via different mechanisms. For example, the rate-determined step of the ORR on Cu–N_2_ is *OH desorption, while that of *OOH is for Cu–N_4_ (Figure 8c). The calculated density of states (DOS) of Cu–N_2_ and Cu–N_4_ indicated that both of them exhibit an excess of d states near the Fermi level (Figure 8e,f), endowing the atomically dispersed Cu–N_x_ moiety with an appropriate interaction with oxygen-containing intermediates and a reduced barrier of ORR catalysis [54].

In contrast to the incomplete d-shells of Fe (3d^6^4s^2^) and Co (3d^6^4s^2^), Mn has a half-filled d-electron shell (3d^5^4s^2^) stable structure which is considered as an inferior active metal to Co and Fe. However, by downsizing Mn onto the atomic scale and tuning the surrounding coordination environment, Mn can be activated to catalyze the ORR due to its d-electrons being tuned to a rational state [33,55]. This may provide valuable and promising guidance to exploit more low-cost and relevant inert metal resources for the active SACs system.

In average, based on current reports, the catalytic activity of diverse SACs in ZAB catalysis primarily follows the order: Fe–N_x_ > Co–N_x_ > Cu–N_x_ > Mn–N_x_. To illustrate the recent advances of SACs in ZABs, Table 1 summarizes the specific activities according to the recent reports.

## 4. Summary and Prospect

In summary, SACs have been introduced in the ZABs field as unique and new frontier catalysts. Due to their maximum atom-utilization efficiency, fully exposed active sites, and the increased ratio of undercoordinated atoms, SACs show exceptional catalytic activity and selectivity in ZABs. This gives SACs great potential as alternatives to noble metal catalysts in ZABs. Among the diverse SACs, Co- and Fe-SACs perform best for the ORR, while more kinds of metal that used to be considered less active catalysts have been exploited as active SAC catalysts for ZABs. Therefore, it is also worth exploring more metal resources for SAC catalysts in ZABs.

However, the single atomization of metal atoms alone is not sufficient to efficiently catalyze the ORR/OER in ZABs and achieve a high power density. To satisfactorily catalyze the reactions in ZABs, a sufficient number of active sites and a high intrinsic activity of the active sites are extremely critical for the catalysts. In addition, even in the same MN_x_ moiety-based SACs, the catalytic activity may be quite different. This can be ascribed to the different atom loading densities on the carbon support and the different intrinsic activity of the isolated active sites. Obviously, the two factors are greatly influence by synthetic methodologies. Furthermore, the density of single-atom sites may change the mechanism for electrochemical reactions. For example, high density single-atom sites may facilitate the formation of adjacent active sites which benefit from the cleavage of the O–O bond and results in a four-electron pathway for the ORR. Therefore, it is extremely critical to take appropriate synthetical strategies to fabricate satisfying single-atom catalysts. To fabricate efficient SACs for zinc–air batteries, the following points may be considered.

(1) Selecting a desired metal-based complex which can be converted to isolated single-atom sites helps achieve a high loading single-atom catalyst. Due to the high surface energy of the isolated atoms, most the current single-atom catalysts maintain very low loading on the support against the migration and agglomeration upon synthesis and application. Because the surface energy sharply increases when the particle size is decreased from the nano scale to atomic scale, low temperature pyrolysis can avoid the agglomeration of metal atoms upon synthesis.

(2) A complex with abundant coordination sites and various hard or soft templates may be an optimized precursor to obtain a high loading of single atoms on the support and provide a high electrochemical active surface area for the catalysts to react. The coordination between the active single atom and the coordination sites exerts a space isolation effect against the agglomeration during high temperature annealing. The high electrochemical active surface area can endow the catalyst with an excellent accessibility to reactions, which facilitates the process of electrochemical reaction.

(3) Binding the active single atom with a foreign atom could be an effective strategy to enhance the catalytic activity of single-atom catalyst. The enhancing active sites tend to lengthen O–O bond lengths, which facilitate the cleavage of O–O bonds and thus achieves a four-electron pathway. Inspired by the previous results, incorporating noble metal atoms such as Pt, Pd, or Au with a noble metal atom may be an effective way to alter their two-electron mechanism into a four-electron mechanism for the ORR. This may extend the application of the cost-effective noble SACs into energy conversion systems such ZABs and fuel cells.

Overall, due to their unique structure and unsaturated properties, SACs have demonstrated their excellent catalytic activity and a promising potential for catalyzing various electrochemical reactions. Additionally, it is believed that their high performance can promote ZABs as a competitive energy conversion system in comparison with other commercial energy devices.

## Figures and Tables

**Figure 1 nanomaterials-09-01402-f001:**
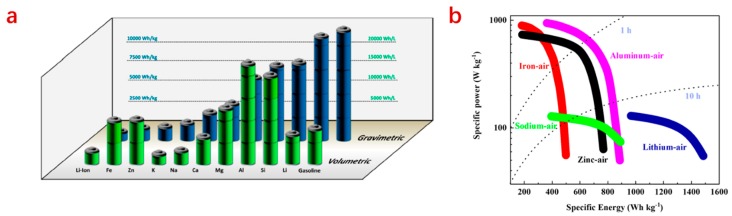
Theoretical energy of various metal–air batteries (excluding oxygen uptake) (**a**) and the Ragone plots for some important and established metal–air batteries (**b**) [11].

**Figure 2 nanomaterials-09-01402-f002:**
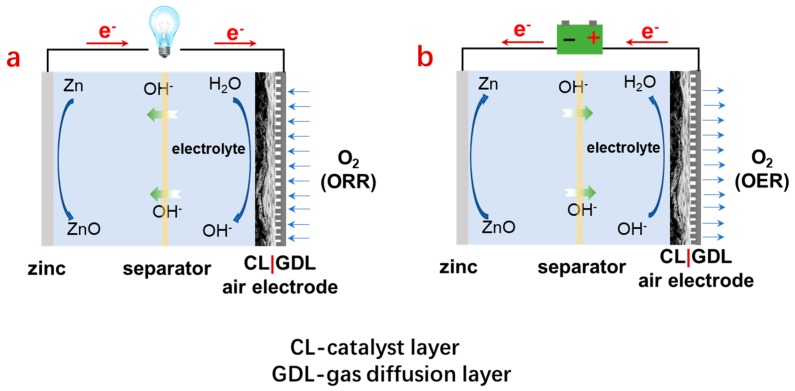
Schematic illustration of configuration of zinc–air batteries (ZABs) and their working principle in the discharge (**a**) and charge (**b**) processes.

**Figure 3 nanomaterials-09-01402-f003:**
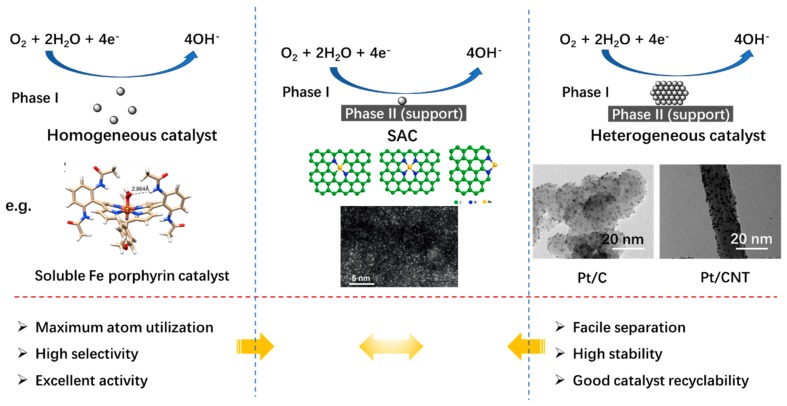
Comparison of diverse catalysis systems for the oxygen reduction reaction (ORR) in alkaline media and the corresponding merits for the corresponding catalysts. The pictures were adopted from [40,41,42,43].

**Figure 4 nanomaterials-09-01402-f004:**
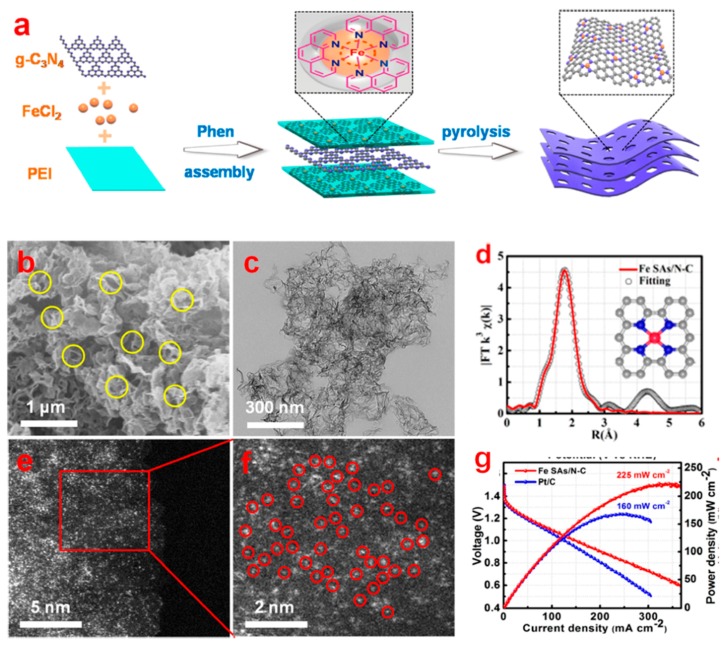
Schematic illustration of the synthetic strategy for single Fe atoms on porous N-doped carbon (Fe-SAs/N–C) (**a**), SEM image (**b**), Transmission Electron Microscope (TEM) image (**c**), Fourier transforms Extended X-ray fine structure (FT-EXAFS) fitting curve and the proposed atomic structure model (**d**), aberration-corrected high-angle annular dark-field scanning TEM (AC HAADF-STEM) images (**e**,**f**), and polarization curves in a zinc–air batteries (**g**) [68].

**Figure 5 nanomaterials-09-01402-f005:**
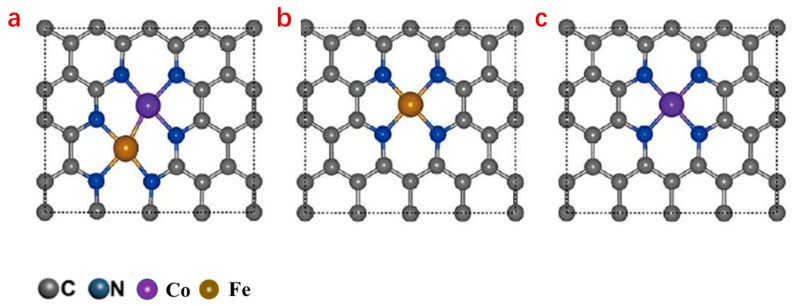
Structure comparison for dual-metal single-atom catalysts (SAC) ((**a**); (Fe, Co–N_x_/C)) and mono-metal SACs ((**b**); Fe–N_x_/C; **c**, Co–N_x_/C) [73].

**Figure 6 nanomaterials-09-01402-f006:**
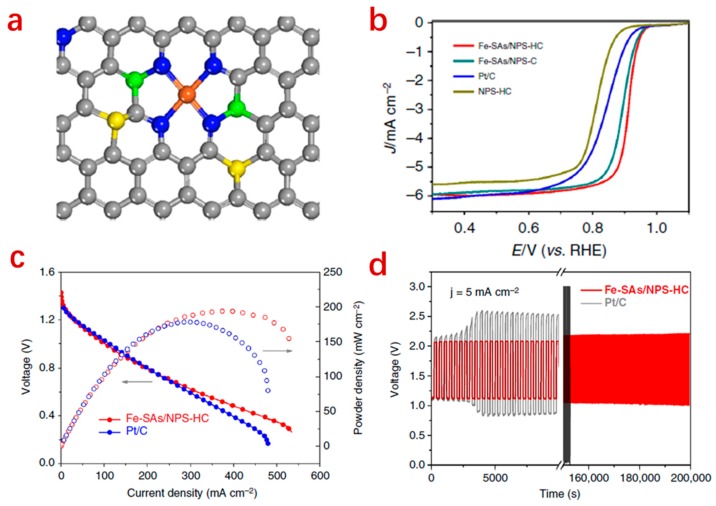
Structure of a ternary doped Fe–N_x_/C catalyst (Single iron atomic sites supported on a nitrogen, phosphorus and sulfur co-doped hollow carbon polyhedron (Fe–SAs/NPS)) (Fe (orange), N (blue), P (green), S (yellow) and C (gray)) (**a**), ORR performance comparison with single-doped and binary-doped analogues (**b**), and performance comparison for ZABs with these SACs and Pt/C catalysts, respectively (**c**,**d**) [78].

**Figure 7 nanomaterials-09-01402-f007:**
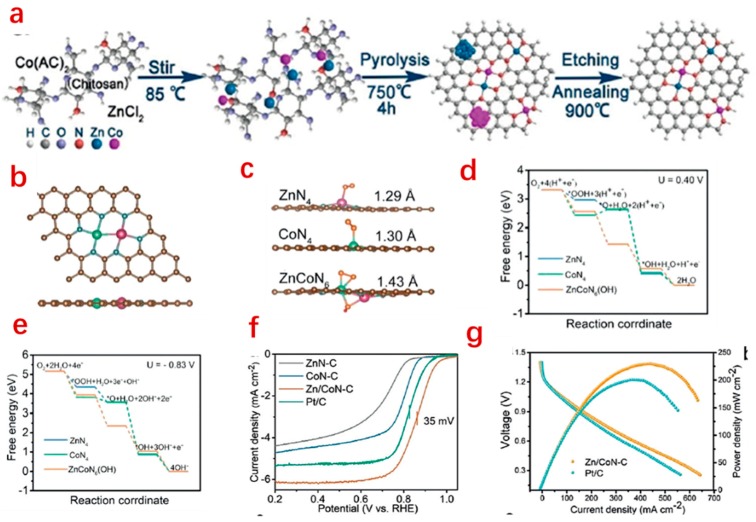
Schematic illustration of the synthesis process of Zn/CoN–C (**a**); the proposed structure of Zn/CoN–C (**b**); O–O bond length after adsorption on the various system (**c**); (brown, blue, yellow, green, purple, and red balls are C, N, O, Zn, Co, and H atoms, respectively) free energy diagrams (**d**,**e**); ORR activity in an alkaline electrolyte (**f**); and zinc–air batteries performance (**g**) [74].

**Figure 8 nanomaterials-09-01402-f008:**
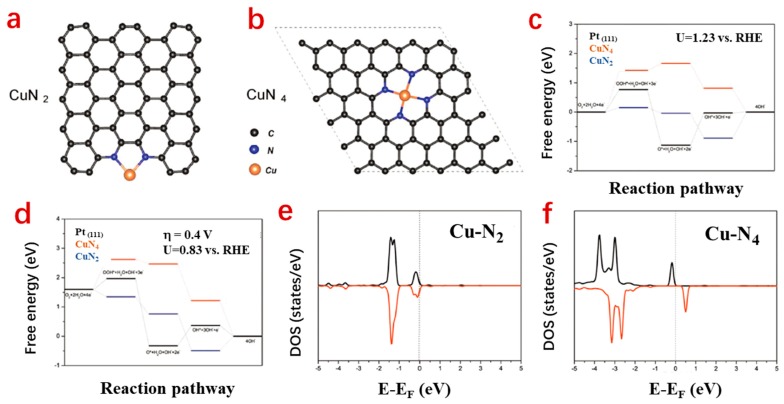
Optimized structures of Cu–N_2_ (**a**) and Cu–N_4_ (**b**); ORR free energy diagrams of CuN_2_ and CuN_4_ (**c**,**d**) at different potentials; local density of states (LDOS) of the d-orbitals of Cu atoms in Cu–N_2_ (**e**) and Cu–N_4_ (**f**) [54].

**Table 1 nanomaterials-09-01402-t001:** Performance of ZABs using the recent representative SACs.

Catalyst	Active Material	ZABs Performance	Ref
Fe-SAs/N–C	FeN_4_ embedded in an N-doped carbon matrix.	The maximum power density is 225 mW cm^−2^. The specific capacity is ∼636 mAh g^−1^. 260 h operation without significant discharge voltage loss.	[67]
Fe–NCCs	Atomic Fe-N_x_ dispersed in a carbon matrix.	The maximum power density is 66 mW cm^−2^. The specific capacity of 705 mAh g^−1^ at 5 mA cm^−2^ and a negligible voltage loss after continuous operation for 67 h.	[53]
SA–Fe/NG	Fe-pyrrolic-N species ON N-doped graphitic carbon.	The maximum power density is 91 mW cm^−2^. A negligible voltage loss after continuous operation for 20 h.	[91]
Fe-SAs/NPS-HC	Single iron atomic sites supported on a nitrogen, phosphorus and sulfur co-doped hollow carbon polyhedron.	The maximum power density is 195 mW cm^−2^. A negligible voltage loss after continuous operation for 55.6 h.	[78]
Fe-SAs/MC	FeN_x_ moiety supported on N-doped mesoporous carbon.	The specific capacity is ∼739 mAh g^−1^ at a discharge current density of 5 mA cm^−2^. A negligible voltage loss after continuous operation for 3 cycles.	[92]
Fe–N-SCCFs	Fe, N-codoped, graphitic simple-cubic carbon frameworks.	The maximum power density is 297 mW cm^−2^. A negligible voltage loss after continuous operation for after 16 h at 10 mA cm^−2^ and 10 h at 50 mA cm^−2^.	[93]
A–Co/r-GO	Atomically-dispersed Co on reduced graphene oxide.	The maximum power density is 225 mW cm^−2^. The specific capacity of 795 mAh g^−1^ and a negligible voltage loss after continuous operation for 50 h.	[94]
Zn/CoN–C	Zn and Co dual metals atoms coordinated by N on a carbon support.	The maximum power density is 230 mW cm^−2^. A negligible voltage loss after continuous operation for 28 h at 5 mA cm^−2^.	[74]
Zn,Co–N_x_-C–Sy	Sulfur (S)-modified Zn and Co–N_x_–C–Sy bimetallic sites embedded in dendritic carbon.	The maximum power density is 150 mW cm^−2^. A negligible voltage loss after continuous operation for 22 h at 5 mA cm^−2^.	[75]
Mn/C–NO	O and N atoms coordinated Mn active sites incorporated within graphene frameworks.	120 mW cm^−2^ at 0.7 V The maximun power density is 120 mW cm^−2^. A negligible voltage loss after continuous operation for 5.6 h at 20 mA cm^−2^.	[55]
S-600	Atomically dispersed Cu–N_x_ moiety in a 3D graphene framework.	The maximum power density is 160 mW cm^−2^. A negligible voltage loss after continuous operation for 10 h at 20 mA cm^−2^.	[54]
FeN_x_–PNC	FeN_x_ moiety on a 2D porous N-doped carbon layer.	The maximum power density is 278 mW cm^−2^. A negligible voltage loss after continuous operation for 40 h at 5 mA cm^−2^.	[49]
FeNPC	N and P co-coordinated Fe atoms in carbon hollow spheres.	233.2 mW cm^−2^ at 0.79 V The maximun power density is 233.2 mW cm^−2^. A negligible voltage loss after continuous charge/discharge for 15 h at 3 mA cm^−2^.	[95]
Meso/micro-Fe–N_x_–CN-30	Meso/microporous FeCo–N_x_-carbon nanosheets.	The maximum power density is 150 mW cm^−2^. A negligible voltage loss after continuous charge/discharge for 28 h at 5 mA cm^−2^.	[96]
S, N–Fe/N/C-CNT	Atomically dispersed Fe-N_x_ on N and S co-doped hierarchical carbon layers.	The maximum power density is 102.7 mW cm^−2^. A negligible voltage loss after continuous charge/discharge for over 100 cycles.	[77]
NGM–Co	Co/N/O tri-doped graphene mesh.	The maximum power density is 152 mW cm^−2^. The specific capacity at 20.0 mA cm^−2^ is ~750 mAh g^−1^. A negligible voltage loss after continuous charge/discharge for 60 h at 2 mA cm^−2^.	[57]
Fe-NSDC	N and S co-doped Fe–N–C species.	The maximum power density is 225.1 mW cm^−2^. The specific capacity at 4 mA cm^−2^ is ~740.8 mAh g^−1^. A long deep cycle life over the last 100 cycles with the charge–discharge overpotential changed from 0.70 V at the 300th cycle to 0.71 V at the 400th cycle, corresponding to 1.4% decrease of voltaic efficiency.	[47]
Fe–Nx–C	Isolated single-atom iron on N-doped carbon frameworks.	The maximum power density is 96.4 mW cm^−2^. The specific capacity is 641 mAh g^−1^. A negligible voltage loss after continuous charge/discharge for 33.3 h at 10 mA cm^−2^.	[97]
Co–N, B–CSs	Boron (B)-doped Co–N–C active sites confined in hierarchical porous carbon sheets.	The maximum power density is 100.4 mW cm^−2^. The specific capacity is 641 mAh g^−1^. A negligible voltage loss after continuous charge/discharge for 14 h at 5 mA cm^−2^.	[51]
NC–Co-SA	Co–N_x_ sites in N-doped porous carbon nanoflake arrays.	The maximum power density of the all-solid-state ZAB is 20 mW cm^−2^. The all-solid state battery showed very stable upon charge/discharge for 2500 min (125 cycles) in its flat state and 2200 min (110 cycles) in its bent state.	[80]
CoN4/NG	CoN_4_ moiety dispersed on N-doped graphitic nanosheet.	The maximum power density is 115 mW cm^−2^. The specific capacity is 730 mAh g^−1^. A negligible voltage loss after continuous charge/discharge for 100 h at 10 mA cm^−2^.	[98]
EA-Co-900	Isolated Co single atoms anchored on N-doped hollow carbon tube.	The maximum power density is 73 mW cm^−2^. A negligible voltage loss after continuous charge/discharge for 100 h at 20 mA cm^−2^.	[99]
Co SA@NCF/CNF	Single Co atoms anchored N-doped carbon flake arrays grown on carbon nanofibers.	The specific capacity is 530.17 mAh g^−1^. A negligible voltage loss after continuous charge/discharge for 90 cycles.	[100]
SCoNC	Monodispersed Co single atoms on a N-doped 2D carbon nanosheets	The maximum power density is 194 mW cm^−2^. The specific capacity is 690 mAh g^−1^ at 10 mA cm^−2^. A negligible voltage loss after continuous charge/discharge for 20 h at 5 mA cm^−2^.	[88]

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
