# Peer review of "Recent Advances in Isolated Single-Atom Catalysts for Zinc Air Batteries: A Focus Review"

_nanomaterials, 2019, doi:10.3390/nano9101402_

Round 1
Reviewer 1 Report
Referee Report
on paper “ Recent advances in the isolated single-atom catalysts for zinc air 3 batteries: a focus review “ (nanomaterials-599524) by authors Weimin Zhang, Yuqing Liu, Lipeng Zhang, Jun Chen submitted to Nanomaterials
This is interesting review. It reports the zinc air batteries which have received much attention due to their theoretically high energy density, excellent safety, and the abundance of zinc resource. Authors focused on the advance application of single-atom catalysts in zinc air batteries in recent years. A main emphasis is then focused on the synthesis and electrocatalytic activity as well as the underlying mechanisms for mono-, and dual-metal based single-atom catalysts in zinc air batteries catalysis. Finally, a prospect is provided, expecting to guide the rational design and synthesis of zinc air batteries. The work done is interesting. However, a few points should be improved. I think that this paper can be published after corresponding addition:
In the review, it is necessary to mention anion-deficient perovskites as promising materials for use as air electrodes, since they have increased ionic conductivity:(1). I.O. Troyanchuk, S.V. Trukhanov, D.D. Khalyavin, H. Szymczak, Magnetic properties of anion deficit manganites Ln0.55Ba0.45MnO3-γ (Ln=La, Nd, Sm, Gd, γ⩽0.37), J. Magn. Magn. Mater. 208 (2000) 217-220. https://doi.org/10.1016/S0304-8853(99)00529-6.
(2). E.V. Antipov, A.M. Abakumov, S.Ya. Istomin, Target-Aimed Synthesis of Anion-Deficient Perovskites, Inorg. Chem. 47 (2008) 8543-8552. https://doi.org/10.1021/ic800791s.
This information should be mentioned in 1. Introduction.
High ionic conductivity is especially pronounced for A-site ordered manganites:(3). S.V. Trukhanov, L.S. Lobanovski, M.V. Bushinsky, V.V. Fedotova, I.O. Troyanchuk, A.V. Trukhanov, V.A. Ryzhov, H. Szymczak, R. Szymczak, M. Baran, Study of A-site ordered PrBaMn2O6-δ manganite properties depending on the treatment conditions, J. Phys.: Condens. Matter 17 (2005) 6495-6506. https://doi.org/10.1088/0953-8984/17/41/019.
(4). N. Hou, T. Yao, P. Li, X. Yao, T. Gan, L. Fan, J. Wang, X. Zhi, Y. Zhao, Y. Li, A-Site Ordered Double Perovskite with in Situ Exsolved Core–Shell Nanoparticles as Anode for Solid Oxide Fuel Cells, ACS Appl. Mat. Interf. 11 (2019) 6995-7005. https://doi.org/10.1021/acsami.8b19928.
This information should be also mentioned in 1. Introduction.
The presented 4 papers should be inserted in References.The paper should be sent to me for the second analysis after the minor revisions.
Author Response
We have revised the manuscript and added changed in the introduction.
All the references have been added as Refs 29-32.
Reviewer 2 Report
This review provides the recent advances in the isolated single-atom catalysts (SACs) for cathode materials in Zn air batteries (ZABs), which required high electrocatalytic activity towards both ORR and OER. The authors focus on only M-Nx/C (M = Fe, Co. Mn, and Cu) catalysts and also emphasize on their synthesis method, catalytic activity, and underlying ORR/OER mechanisms. This review is well organized and will be useful resource for further design and development of SACs for ZABs. I think this review can be accepted after minor revision. Comments are listed as follows.
There is a text box (S20-12cm) in Figure 4a. Does it have any meaning? Please provide or label the elemental identification of each atom color in Figure 4. Figure 2: please revise “soluable” to “soluble”. Figure 5b is overlapped by Figure 5b. “E/V(vs. RHE) was partially disappeared. Font size in Figure 7c, 7d, 7e, and 7f is quite small. Can the authors increase the font size? Format of Table 1 does not follow the journal template. Please correct.
Author Response
Thanks very much for your comments and valuable suggestions.
The text box (S-20-12cm) was due to the system and typing error, and has been removed.
In Figure 2, "soluable has been corrected with "soluble".
Figure 5b and Figure & have been redrawn as suggested.
Table 1 has been reformatted based on the journal template.
Reviewer 3 Report
Thank you for invitation to review the submitted draft. The review can be accepted for publications but require major revision.
Authors should have given an overview and advantages of ZABs with Li Na and K air batteries. How, the low cell potential of ZAB can competete with alkali metal systems. Authors have discussed about the advantage of having double metals however, it would have been great if they can provide clear picture of reaction mechanism during ORR and OER specifically, mechanism for OER isn't clear in literature as well. It would have been essential if authors can provide the current status of ZAB by comparing it with other metal-air battery system such as Li-O2, Na-O2 hybrid Li-O2 and hybrid Na-O2 atleast in Ragone plot. Some important recent references related to air battery should be included https://doi.org/10.1038/s41563-019-0286-7; Journal of Materials Chemistry A, 2018, 6 (47), 24459-24467 etc.
Author Response
Changes have been made in introduction (page 2) to give a proper overview and advantages of ZABs with Li, Na, and K-air batteries.
In comparison with mono-metal SACs, few reports have been published on dual-metal SACs with proper mechanism investigation. Therefore, at current stage, we would prefer not to give a mechanism picture of dual-metal SACs, in order to avoid scientific confusion and misleading. The corresponding discussion has been added in page 9.
Following the reviwer' comment, we've modified the Ragone plots in FIgure 1b to demonstrate a proper comparison between ZAB and other metal-air system. However, only a few papers report hybrid system, which make us be unable to draw a confident Ragone plot specific for the hybrid system. Therefore, at current stage, we did not add this.
Related references have been added as Refs 9 and 20.